# C1q/TNF-Related Protein 3 Prevents Diabetic Retinopathy via AMPK-Dependent Stabilization of Blood–Retinal Barrier Tight Junctions

**DOI:** 10.3390/cells11050779

**Published:** 2022-02-23

**Authors:** Zheyi Yan, Chunfang Wang, Zhijun Meng, Lu Gan, Rui Guo, Jing Liu, Wayne Bond Lau, Dina Xie, Jianli Zhao, Bernard L. Lopez, Theodore A. Christopher, Ulhas P. Naik, Xinliang Ma, Yajing Wang

**Affiliations:** 1Department of Emergency Medicine, Thomas Jefferson University, Philadelphia, PA 19107, USA; yanzheyi1127@163.com (Z.Y.); zheyizheyiyan@hotmail.com (C.W.); dr.zhijunmeng2019@gmail.com (Z.M.); ganlu@wchscu.cn (L.G.); rui.guo@sxmu.edu.cn (R.G.); jingliu@sxmu.edu.cn (J.L.); waynebond.lau@jefferson.edu (W.B.L.); dina.xie@outlook.com (D.X.); jianli.zhao@jefferson.edu (J.Z.); bernard.lopez@jefferson.edu (B.L.L.); theodore.christopher@jefferson.edu (T.A.C.); xinliang.ma@jefferson.edu (X.M.); 2Department of Medicine, Cardeza Center for Hemostasis, Thrombosis, and Vascular Biology, Cardeza Foundation for Hematologic Research, Thomas Jefferson University, Philadelphia, PA 19107, USA; ulhas.naik@jefferson.edu

**Keywords:** CTRP3, iBRB, tight junction proteins, permeability, diabetic retinopathy

## Abstract

**Background** The impairment of the inner blood–retinal barrier (iBRB) increases the pathological development of diabetic retinopathy (DR), a severe complication in diabetic patients. Identifying approaches to preserving iBRB integrity and function is a significant challenge in DR. C1q/tumor necrosis factor-related protein-3 (CTRP3) is a newly discovered adipokine and a vital biomarker, predicting DR severity. We sought to determine whether and how CTRP3 affects the pathological development of non-proliferative diabetic retinopathy (NPDR). **Methods** To clarify the pathophysiologic progress of the blood–retinal barrier in NPDR and explore its potential mechanism, a mouse Type 2 diabetic model of diabetic retinopathy was used. The capillary leakage was assessed by confocal microscope with fluorescent-labeled protein in vivo. Furthermore, the effect of CTRP3 on the inner blood–retinal barrier (iBRB) and its molecular mechanism was clarified. **Results** The results demonstrated that CTRP3 protects iBRB integrity and resists the vascular permeability induced by DR. Mechanistically, the administration of CTRP3 activates the AMPK signaling pathway and enhances the expression of Occludin and Claudin-5 (tight junction protein) in vivo and in vitro. Meanwhile, CTRP3 improves the injury of human retinal endothelial cells (HRMECs) induced by high glucose/high lipids (HG/HL), and its protective effects are AMPK-dependent. **Conclusions** In summary, we report, for the first time, that CTRP3 prevents diabetes-induced retinal vascular permeability via stabilizing the tight junctions of the iBRB and through the AMPK-dependent Occludin/Claudin-5 signaling pathway, thus critically affecting the development of NPDR.

## 1. Introduction

Despite improvements in clinical patient care, risk factor identification and public health awareness of diabetes are still lacking well-noted research. A large population of diabetes patients progress to diabetic retinopathy (DR), a problematic complication of diabetes, causing blindness, especially in developed countries [1]. Diabetic retinopathy is considered a microvascular complication of endothelial dysfunction, characterized by damage of the blood–retinal barrier (BRB), and neovascularization [2]. The BRB is composed of two distinct barriers; the outer BRB, consisting of retinal pigment epithelium, and the inner BRB (iBRB), which is formed by the tight junctions between neighboring retinal capillary endothelial cells, and regulates transportation across retinal capillaries. Notably, the loss of iBRB occurs in the early stages of diabetes [3,4] (non-proliferative stage, NPDR) and leads to the development of clinically significant DR [5,6,7]. Determining the novel targets for NPDR not only helps for early DR diagnostics but is also helpful for formulating preventive strategies against advancing late stage DR.

The C1q/tumor necrosis factor-related proteins (CTRPs) are newly identified adiponectin paralogs consisting of 15 (CTRP1-CTRP15) subtypes [8]. Recent studies reported that, regarding the relationship between CTRPs and diabetes, CTRP3 particularly had been shown to exhibit anti-diabetic, anti-inflammatory [9,10,11,12], and metabolic regulatory effects in multiple tissues, including the vasculature [13]. Notably, we have recently demonstrated that serum CTRP3 is associated with DR, serving as a novel biomarker for DR severity [12]. However, it has never been previously investigated whether CTRP3 regulates diabetes-induced iBRB dysfunction in NPDR and how it modulates retinal vascular endothelial cells’ pathophysiological process.

The aims of the current study were: (1) to determine whether CTRP3 modulates the barrier pathophysiological process of iBRB in NPDR, mitigating diabetes-induced retinal vascular permeability, and if so, (2) to determine the responsible mechanisms involved.

## 2. Methods

### 2.1. HFD/STZ Type 2 Diabetes (T2D)-Induced DR Animal Model

All experiments were performed in adherence to the National Institutes of Health Guidelines on the Use of Laboratory Animals and were approved by the Thomas Jefferson University and Shanxi Medical University IACUC Committee on Animal Care (0077300946). In order to establish a Type 2 diabetes (T2D)-induced DR model, adult C57BL/6J mice were fed a high-fat diet containing 60% energy from fat (D12492; Open Source Diets) combined with streptozotocin (STZ) (Sigma Chemical, St. Louis, MO, USA) [14]. The mice were fed HFD for 12 weeks to induce obesity, characterized by abnormal glucose tolerance and insulin resistance. The age-matched nondiabetic mice were fed a standard diet (SD-diet). Then, mice were subjected to STZ (50 mg/kg body weight, once) at 12 weeks of HFD [14] followed by 8 more weeks of HFD to establish a DR model (Appendix A). In addition, mice fed a standard diet as controls were given an injection of an equivalent volume of citrate buffer. Fasting blood glucose level was examined using a RocheAccu-Chek™ blood glucose monitoring system at 12 weeks of HFD. Mice were considered diabetic when blood glucose exceeded 12 mmol/L. The biologically active globular domain of CTRP3 (gCTRP3, 0.5 μg/g/d) or double-distilled water was administered to HFD/STZ-T2D mice and the age-matched non-diabetic mice, respectively, via peritoneally implanted osmotic pumps during the last 2 weeks of HFD (Appendix A). Meanwhile, to inhibit AMPK activation, 100 µL (20 mg/mL) Compound C (Sigma Chemical, St. Louis, MO, USA) was intraperitoneally injected into diabetic mice daily during the last 2 weeks.

### 2.2. Glucose and Insulin Tolerance Assay

To assess glucose tolerance, mice were intraperitoneally injected with D-glucose (1.5 g/kg) after fasting. The venous blood was collected 30 min before (time 0) and after injection at 0, 15, 30, 60, and 120 min. Per the manufacturer’s instructions, the blood glucose was measured using a Roche Accu-Chek™ glucose monitoring system. To assess insulin tolerance, a single dose of Novolin R regular insulin (0.5 unit/kg) was intraperitoneally administered to the mice after fasting for 4 h with free access to water. The blood glucose level was measured as described above.

### 2.3. Capillary Permeability Assay In Vivo

To observe capillary permeability in vivo, the T2D-DR mice and the age-matched non-diabetic mice were anesthetized. Evans Blue (EB) dye (25 mg/kg, 0.5% in PBS) was injected via the tail vein. At 5 min after injection, the mice were euthanized. Mouse ear tissue was collected with an 8 mm skin punch and fixed in 10% formamide at 56 °C for 48 h. The quantification of Evans blue in the tissues was determined by the optical density assessment on Image J (NIH) [15]. To assess CTRP3′s role in ear capillary permeability, gCTRP3 (0.5 μg/g/d) or vehicle was administered via peritoneally implanted ALZET osmotic pumps (Cupertino, CA, USA) for 2 weeks to T2D-DR mice and the age-matched non-diabetic mice, respectively.

### 2.4. Retinal Capillary Leakage Assay In Vivo

Retinal capillary leakage assay was analyzed by previously reported methods with modifications [15,16,17,18]. Briefly, the mixture of tetramethylrhodamine isothiocyanate conjugate Concanavalin A (TRITC-ConcanavalinA) (indicating vascularity, 25 μg/g, Cat#C860, Thermo Fisher Scientific, MA, USA) and FITC-dextran (indicating vascular leakage, 50 μg/g, Cat#46945, Sigma-Aldrich, St. Louis, MO, USA) were injected into the superior vena cava. Then, the mice were euthanized at 15 min following this, and retina tissues were obtained. The retina was flatly mounted and used for examining the retinal vascular pattern. Images were obtained with a Nikon EclipseE800 confocal microscope (Nikon). Five images for each eye (full view) were taken. The mean intensity of TRITC-ConcanavalinA and FITC-dextran was determined by Image J software (NIH).

### 2.5. Cell Culture and Treatments

Human retinal microvascular endothelial cells (HRMECs) (Cat# cAP-0010, Angio-Proteomie, Boston, MA, USA) were plated on six-well plates and cultured at 37 °C in a 5% CO_2_ incubator. Upon reaching 100% confluence, HRMECs were randomized to receive one of the following treatments: (1) normal glucose normal lipids (NGNL, 5.5 mM D-glucose/19.5 mM L-glucose,) receiving vehicle or gCTRP3 (0.3, 1, and 3 μg/mL, Cat# 00082-01; Aviscera Bioscience, Santa Clara, CA, USA) for 24 h; (2) high glucose/high lipids (HGHL, 25 mM D-glucose/250 μM palmitates) [19], receiving vehicle or gCTRP3 treatment (0.3, 1, and 3 μg/mL) for 24 h; 25 mM L-glucose plus fatty acid-free BSA was used as an osmotic control [20].

### 2.6. iBRB Permeability Assay In Vitro

The permeability of the endothelial membrane was assessed by the passage of FITC-dextran through the HRMEC monolayer [21]. To measure endothelial barrier function in iBRB, an in vitro trans-well endothelial permeability assay (Sigma-Aldrich, MA, USA) was performed. Briefly, after the HRMEC monolayer was built, the culture medium was pre-treated with gCTRP3 (3 μg/mL) for 15 min, followed by NGNL/HGHL challenge for 24 h. FITC-dextran (20 μg/mL) was added to the upper chamber for 20 min. The fluorescence signal in the lower chamber was determined at 450 nm using a Bio-Rad 450 microplate reader (Bio-Rad, Hercules, CA, USA).

To investigate the continuous real-time monitoring of the HRMEC monolayer migration and cell–cell junctions in vitro, we conducted xCELLigence electrical conductivity assays [22,23]. HRMECs (3.0 × 10^4^ cells) were seeded onto a 16-well E-plate. After equilibration, plates were inserted into the xCELLigence station, and the baseline impedance was measured to ensure that all wells and connections were working within acceptable limits (Appendix A). Cell index curves were determined by the xCELLigence RTCA System (Roche, RTCA DP Station) per manufacturer instruction.

### 2.7. Western Blot Analysis

HRMECs or tissues were lysed by lysis buffer (Cell Signaling, Topsfield, MA, USA). Total protein (50 μg) was loaded on Bio-Rad 4–20% gel system, transferred to PVDF membrane, and immunoblotted with one of the following primary antibodies: Serine473 phosphorylated Akt (Cat#4060, Cell Signaling), total Akt (Cat#4691,Cell Signaling), Threonine172 phosphorylated AMPK (Cat#2535, Cell Signaling), total AMPK (Cat#5832, Cell Signaling), Acetyl-CoA Carboxylase (Cat #3662, Cell Signaling), Phospho-Acetyl-CoA Carboxylase (Cat #11818, Cell Signaling), Claudin-1 (Cat#13995, Cell Signaling), zo-1 (Cat#13663, Cell Signaling), Claudin-5 (Cat#35-2500, Thermo-Fisher Scientific), Occludin (Cat#71-1500, Thermo-Fisher Scientific), and ß-Actin (Cat#SC-4778, Sigma). The secondary antibody was obtained from Cell Signaling Company. The membrane was exposed to Super-Signal Reagent (Pierce, IL, USA), and imaged on a Bio-Rad ChemiDoc Touch station (Bio-Rad).

### 2.8. Cellular Immuno-Fluorescence Staining

Cells were fixed with 1% paraformaldehyde for 10 min at room temperature. The cells were incubated with primary antibodies against anti-Claudin 5 (Cat#35-2500, Thermo-Fisher Scientific) and Occludin (Cat#71-150, Thermo-Fisher Scientific). After overnight incubation at 4 °C, the cells were incubated with fluorescein-labeled secondary antibodies. The nuclei were counterstained with 4′,6-diamidino-2-phenylindole (DAPI). Micrographs of all immunostains were acquired by an Olympus BX51 Fluorescence Microscope and an Olympus DP72 image system. Quantification of fluorescence intensity at the cell boundary was performed with Image J (NIH).

### 2.9. Measurement of Serum and Retinal Globular Domain CTRP3 (gCTRP3)

Animal retina and serum were collected and stored at −80 °C. Frozen retinas were homogenized, centrifuged for 10 min at 13,000× *g* at 4 °C, and the supernatants were collected. The serum was centrifuged for 15 min at 3000× *g* at 4 °C, and the supernatants were collected. gCTRP3 was measured using a commercial ELISA kit (Cat: SK00082-03-100, Aviscera Bioscience, Santa Clara, CA, USA) per the manufacturer’s instruction. The minimum detectable level of CTRP3 was 16 ng/mL (intra-assay coefficient of variability (CV): 4–6%; inter-assay CV: 8–10%).

### 2.10. Tube Formation Assay

To examine the impact of CTRP3 on the tube formation of HRMECs, we used a u-slide angiogenesis system (ibidi). Briefly, the plate was coated with Matrigel (growth factor reduced, BD Biosciences). Upon gel solidification, HRMECs (1 × 10^4^ cells) were seeded onto pre-coated Matrigel. Then, gCTRP3 (0.3, 1, and 3 μg/mL) was added and then the insert was removed. After overnight incubation, the capillary-like structures of HRMECs were photographed with a microscope and analyzed via Image J software (NIH). Endothelial cell basal medium was replaced with NG medium or HGHL medium for conditioned medium experiments.

### 2.11. Small Interfering RNA Transfection

HRMECs were transfected by an siPORTER siRNA transfection kit (Cat#AM4503, Ambion) per manufacturer’s protocol with siRNA duplex against AMPK-α1 and universal negative control (Santa Cruz). Briefly, HRMECs were plated on six-well plates. After HRMECs reached 80% confluence, siRNA was applied to each well (final concentration 50 nM). Protein suppression was evaluated by Western blot 72 h later.

### 2.12. Statistical Analysis

Data were analyzed with GraphPad Prism-7 statistics software (La Jolla, CA, USA). All values are presented as the mean ± SD of independent experiments. To examine differences between the two groups, the unpaired Student t-test was performed. For multiple groups, one-way ANOVA was conducted across all investigated groups. The Tukey post hoc tests confirmed where statistically significant differences existed between groups. *p* values less than or equal to 0.05 were considered statistically significant.

## 3. Results

### 3.1. HFD/STZ-Induced T2D Increased Microvascular Permeability

The characteristics of DR are illustrated as Appendix A. When compared to ND-fed mice, HFD-fed mice exhibited a significant increase in body weight (Figure 1B). Meanwhile, the glucose tolerance and insulin tolerance assay revealed that mice exhibited abnormal glucose tolerance and insulin resistance when animals were fed with HFD for 12 weeks (Figure 1C,D). However, no significant difference was observed in body weight between the control-treated and HFD/STZ-induced T2D group at 20 weeks (Figure 1E). Merged confocal images of the retinal flat mounts showed increased vascular leakage after HFD/STZ treatment, demonstrating that the diabetic retinopathy model had been established (Figure 1G). Ear capillary permeability assay showed Evans blue dye extravasation (Figure 1F), suggesting that ear microvascular permeability had increased in this DR model.

### 3.2. CTRP3 Exerted a Protective Effect on Capillary Leakage and Retinal Capillary Leakage in the T2D Model

Among the CTRPs investigated in diabetic patients with microvascular disease, CTRP3 reduction is most significantly associated with DR [12]. With this supportive evidence, we further explored whether CTRP3 plays an important role in the prevention of the pathological process of DR. Two experiments were performed. gCTRP3 or vehicle was administered via peritoneally implanted osmotic pumps for 2 weeks to HFD/STZ mice and the age-matched non-diabetic mice. Concentrations of gCTRP3 in serum and retina were decreased in diabetic mice when compared to control. Administration of gCTRP3 (0.5 μg/g/d) replenished gCTRP3 levels (Figure 2A–C). As capillary leakage is a golden standard reflecting microvascular damage, we first performed the Miles permeability assay to identify the gCTRP3 protective effect. Although administration of gCTRP3 had no significant effect on body weight, blood glucose levels, and insulin sensitivity (Figure 1C–F), it markedly prevented extravasation of Evans blue dye (Figure 1A,B). Therefore, these data suggest that gCTRP3 treatment significantly preserved the capillary barrier function in the T2D model. Second, to further illustrate the effect of CTRP3 specifically on retinal vascular permeability, we evaluated retina vascular extravasation after administering CTRP3 to animals with diabetic retinopathy. As illustrated in Figure 2A,C, retina vascular extravasation (green color) was increased in the T2D model. However, it was markedly decreased after CTRP3 was administered, suggesting that CTRP3 effectively reduced retinal capillary leakage. In addition, no neovascularization was observed, indicating that gCTRP3 did not significantly promote angiogenesis (Figure 2A,B).

### 3.3. CTRP3 Protected Barrier Function of the Inner Blood–Retinal Barrier (iBRB) against HGHL-Induced Impairment

Retinal neovascularization is a severe complication in diabetic retinal pathology. To directly investigate whether CTRP3 affects angiogenesis in the early stage of diabetes, we determined the effect of CTRP3 upon tube formation in cultured HRMECs. HRMECs were cultured in NGNL or HGHL medium for 24 h with vehicle or gCTRP3 pre-treatment. CTRP3 had no significant effect on tube formation (Figure 3A,B). Since the inner blood–retinal barrier (iBRB) is a structure of tight junctions among HRMECs preventing retinal capillary leakage, we performed three serial experiments to further clarify whether CTRP3 could protect the integrity and function of the iBRB. First, we utilized in vitro xCELLigence electrical conductivity assays to evaluate the cell–cell junctions of the iBRB. The cell index curve rose sharply and then stabilized, which suggested that iBRB was formed and functional (Figure 2A). As illustrated in Figure 3C, gCTRP3 addition alone did not affect the permeability of the iBRB. However, when CTRP3 was added to HRMECs, followed by HGHL administration, iBRB permeability was successfully prevented by gCTRP3 treatment (Figure 3C,D). Second, we performed an in vitro endothelial trans-well permeability assay. HRMECs’ monolayer (iBRB) permeability was determined by measuring the FITC-dextran signal (iBRB permeability indicator) on the endothelial monolayer as indicated in Figure 2B. The iBRB permeability was significantly increased when challenged with HGHL, whereas the HGHL-induced increase was successfully blocked by gCTRP3 pre-treatment (Figure 3E). Thirdly, we performed immunofluorescence staining for tight junctions with Occludin and Claudin-5 in HRMECs to determine whether gCTRP3 can protect the integrity of iBRB from HGHL-induced damage. HGHL treatment disrupted endothelial cell-to-cell junctions. This disruption of junctions was effectively prevented by gCTRP3 pre-treatment (Figure 3F–H). These results suggested that gCTRP3 maintained the barrier function of iBRB, preventing HGHL-induced permeability.

### 3.4. CTRP3 Increased Occludin and Claudin-5 Expression and Protected iBRB from HGHL-Induced Impairment

To identify the molecular mechanisms underlying the effects of CTRP3 on the iBRB protection, we screened the expression of iBRB related tight junction proteins, including Claudin-1, Zonula occludens-1 (ZO-1), Claudin-5, and Occludin in HRMECs. As indicated in Figure 4A–E, pre-treatment with gCTRP3 increased Occludin and Claudin-5 expression in a dose-dependent manner and successfully prevented the HGHL-induced suppression of Occludin and Claudin-5 levels. gCTRP3 alone did not cause alteration of tight junction protein expression. Taken together, these results provide clear evidence that gCTRP3 increased Occludin and Claudin-5 expression and protected the integrity of iBRB from HGHL-induced damage.

### 3.5. CTRP3 Increased Occludin and Claudin-5 (Tight Junction Protein) Expression in HRMECs via AMPK Activation In Vitro

Next, we screened metabolism and survival related signaling pathways to determine the upstream signaling molecules mediating the expression of Occludin and Claudin-5. Although several studies have reported that the AMPK signal is suppressed in diabetes, [24] whether the protective role of CTRP3 on iBRB’s tight junction observed here in this study is associated with AMPK is unclear. CTRP3 treatment significantly increased AMPK phosphorylation. However, Akt was not significantly activated by CTRP3 in HRMECs (Figure 5A,B). In addition, with AMPK-silenced HRMECs, followed by gCTRP3 pre-treatment (3 μg/mL, 15 min) and HGHL administration, we identified that gCTRP3 failed to block the HGHL-induced decrease of Claudin-5 and Occludin expression (Figure 5C,D), suggesting that AMPK activation is required for gCTRP3′s regulation in Occludin and Claudin-5 expression. Meanwhile, the activation of acetyl-CoA carboxylase (AMPK’s direct target) was evaluated. CTRP3 successfully restored HGHL inhibited-ACC phosphorylation (Figure 5C,D). In order to determine AMPK’s role in CTRP3’s influence on cell-to-cell junctions of iBRB, we utilized an xCELLigence electrical conductivity assay. AMPK silence effectively blocked the protective effect of gCTRP3 on HGHL-induced iBRB disruption (Figure 5E,F). Taken together, these results provided clear evidence that CTRP3 increases Occludin and Claudin-5 tight junctions protein expression and exerts its iBRB- protection effect via AMPK-dependent signaling.

### 3.6. CTRP3-Induced Increase of Occludin and Claudin-5 Is Abolished after Inhibition of AMPK In Vivo

To confirm whether CTRP3 protects iBRB’s tight junctions via AMPK signaling in vivo, 10 mg/kg of Compound C (an AMPK inhibitor) was intraperitoneally injected into diabetic mice daily for 2 weeks. We then assessed pAMPK, AMPK, pACC, ACC, ZO-1, Occludin, Claudin-1, and Claudin-5 expression in each group. Consistent with in vitro data, DR inhibited AMPK and ACC activity and Occludin and Claudin-5 expression. Administration of gCTRP3 reversed those changes in the retinal tissue in the DR model. However, these effects were blocked by Compound C administration (Figure 6A–C). These results suggest that CTRP3 maintained the barrier function of iBRB in DR by preventing the diabetes-induced Occludin and Claudin-5/tight junctions protein disruption, via AMPK-dependent signaling.

## 4. Discussion

In the current study, we report that AMPK activation in the retina was inhibited in diabetes-induced NPDR, consequently suppressing the crucial molecules Occludin and Claudin-5 in iBRB. However, CTRP3 preserves iBRB function in NPDR by preventing diabetes-suppressed Occludin and Claudin-5 (tight junction protein) expression in an AMPK-dependent manner (Figure 6D). This study suggests that CTRP3 may have therapeutic potential to the diabetic retina by decreasing vascular permeability, and preventing CTRP3 reduction is a promising therapeutic approach during the management of DR.

Despite years of ongoing scientific investigation, the fundamental phenotypic properties governing the increased incidence of diabetes and its complications, especially in DR, remain largely unknown. DR development involves two stages, NPDR and PDR. iBRB injury has been implicated in the initiation and progress of retinal complications of diabetes, particularly during NPDR. Plasma leakage and fluid retention are seen in various tissues of diabetic patients or animals at the early stages of the disease (NPDR), before structural microangiopathy can be detected (PDR). CTRP3, as the critical regulatory factor of adipose-derived hormones and cytokines (adipokines and adiponectin paralogs) in diabetes, has gained recognition [8,25,26]. We have recently demonstrated that circulating CTRP3 may serve as a valuable biomarker in the screening of diabetic retinopathy in patients, as it is inversely associated with DR severity [12]. Diabetes is a group of metabolic diseases characterized by hyperglycemia resulting from defects in insulin action. However, this study demonstrated that CTRP3 did not present the metabolic regulatory effect of Type 2 diabetes. At the same time, it prohibited capillary vascular leakage in the late stage of the diabetic model. CTRP3 may therefore have a clinical application to NPDR, a pathology requiring advancements in treatment options. CTRP3 has been reported to exert anti-apoptotic effects [27], and attenuate diabetes-related injuries [28,29] including the inflammatory processes of DR [19], suggesting the therapeutic potential of CTRP3 for the treatment of DR. To test this hypothesis, we utilized the globular domain isoform of CTRP3 (gCTRP3) to investigate its effect on the early stage of diabetic retinopathy (NPDR). gCTRP3 can be generated via the proteolytic cleavage of full-length CTRP3 (fCTRP3), which functions as the active isoform [30]. Although retinal neovascularization is a severe complication in DR, in this study, we focused on CTRP3′s effect on the early stage of diabetic DR (NPDR), when neovascularization did not occur, but CTRP3 can ameliorate the leakage of iBRB as manifested in Figure 2A,B with FITC-Dextran tracing. In addition, we demonstrated that retinal neovascularization was not observed with TRITC-ConcanavalinA tracing. Meanwhile, the tube formation experiment demonstrated that CTRP3 did not directly increase angiogenesis on the cellular level. Therefore, we have confirmed that CTRP3 did not affect retinal angiogenesis in vivo and in vitro, but prevented the permeability increases of the iBRB.

Next, we demonstrated that CTRP3 significantly reduced not only capillaries’ leakage of the retina but also capillaries leakage on micro grade vessels in vivo. Specifically, we have observed that extravasation of Evans blue dye was significantly increased in DR mice, a pathologic alteration markedly inhibited by CTRP3. These results indicate that CTRP3 treatment protects barrier integrity and preserves the capillary junctions from diabetes-induced injury.

Retinal neovascularization is a severe complication in DR. The effect of CTRP3 upon angiogenesis remains controversial [31]. The TRITC-ConcanavalinA/FITC-dextran permeability assay that we utilized in the study is a method to present vascular structures with high resolution and can provide solid evidence on retinal capillary permeability and angiogenesis in the retina. However, based on the in vivo retinal vascularity examination results, we observed that CTRP3 had no significant effect on retinal neovascularization in this early stage of the diabetic DR model (NPDR). It is worth noting that, in our pilot experiments, CTRP3 failed to exert a significant protective effect in the diabetes-induced severe DR mice model (PDR model) (18 weeks after diabetes onset), suggesting that early treatment with CTRP3 is crucial.

Third, we report the protective effect of CTRP3 against HGHL-induced inner blood–retinal barrier (iBRB) impairment. iBRB, as a critical barrier in developing novel therapeutics for DR, contains tight-junctions (TJ’s) among adjacent endothelial cells lining the fine capillaries of the retinal [32,33]. TJs are comprised of membrane-spanning proteins (Claudins, Occludin, and Junctional adhesion molecules) which interact with cytoplasmic proteins (ZO-1) [34,35]. gCTRP3 did not affect iBRB under normal circumstances. However, CTRP3 preserves Occludin and Claudin-5 levels in diabetic HRMECs. Furthermore, consistent with previous clinical studies [36,37], we report that CTRP3 significantly increased AMPK phosphorylation, and that its mechanism is through regulating the levels of Occludin and Claudin-5 in iBRB by CTRP3. These in vitro results were further confirmed by an in vivo DR model, by analysis of tight junction proteins in the retinal tissue. It should indicate that, although multiple AMPK activators, such as metformin, are currently available, CTRP3 has a clear advantage. It is worth noting that CTRP3 does not influence animal weight, blood glucose level, or OGTT. Therefore, CTRP3′s protective effect against the diabetic DR process is independent of glucose regulation. More importantly, metformin is a chemical AMPK activator. Its dose must be precisely titrated to achieve optimal AMPK activation, as overactivation of AMPK may have an adverse effect. In contrast, CTRP3 is an endogenous molecule with AMPK activation capability. A replacement strategy to restore CTRP3 to its average level in the diabetic model (as utilized in the current study) is more desirable.

There are several limitations in the current study. First, we used an HFD/STZ-induced T2D model. Whether CTRP3 may also be protective in a genetic T2D model remains unknown [38,39]. Second, cell type and stimuli conditions may be responsible for the variant effects we observed when compared with other endothelial cell-related studies. In addition, we recognize that the pathophysiology of DR is highly complex and multifactorial. It involves the activation of several interrelated pathways, including increased oxidative stress, increased pro-inflammatory mediators, and induced VEGF secretion. These pathologic alterations all occur on a background of the various metabolic derangements inherent to DM [40], and there is more to learn in these complex interactions. Some questions remain unanswered and need to be further explored. These include how the AMPK signal regulates Occludin and Claudin-5 expression, and whether AMPK regulates their phosphorylation and/or intracellular translocation, especially Claudin-5′s nucleus behaviors and translocation.

## 5. Conclusions

Our results identify CTRP3 as a prominent adipokine, exerting iBRB protection through upregulating Occludin and Claudin-5 in DR in an AMPK-dependent manner. Thus, CTRP3 may serve not only as a biomarker as we previously reported [12] but also as a promising preventive and therapeutic mean for DR and associated complications of T2D.

## Figures and Tables

**Figure 1 cells-11-00779-f001:**
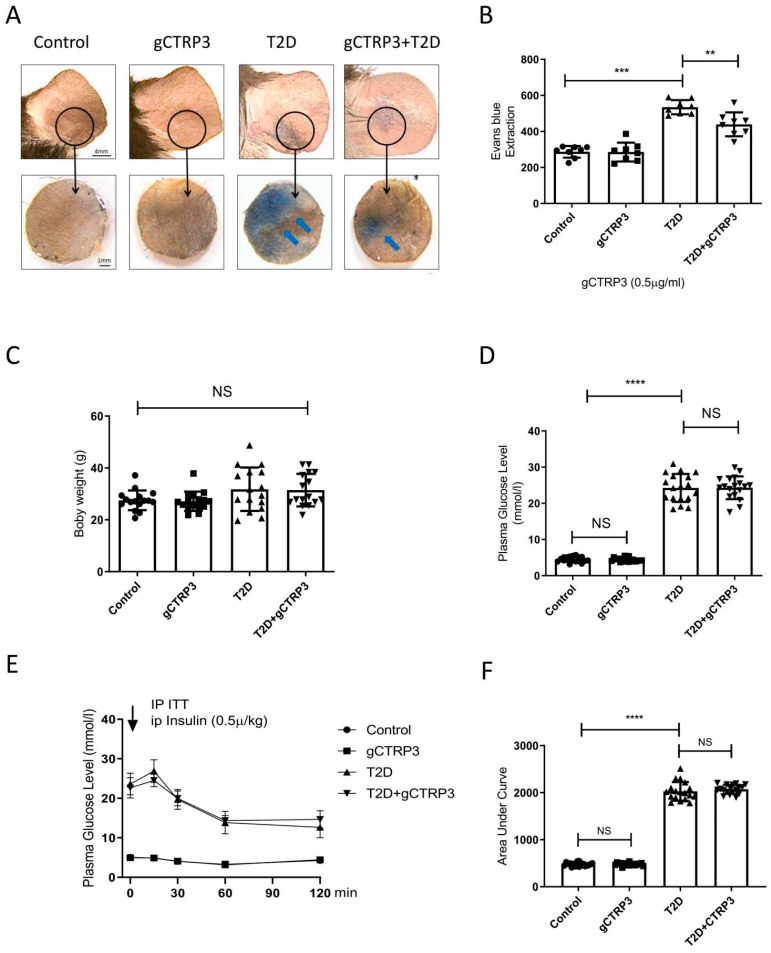
gCTRP3 ameliorated T2D-induced capillary leakage. (**A**), Representative images are showing that gCTRP3 (0.5 μg/g/d) reduced Evans blue dye leakage to dermal adjacent tissue in T2D mice. (**B**), Quantification with column graph analysis (*n* = 8–10). (**C**), gCTRP3 failed to affect body weight, blood glucose levels (**D**), or insulin resistance (**E**,**F**) in the diabetic retinopathy group when compared with Sham. All data are shown as means ± SD. *n* = 18 for each group. ** *p* < 0.01, *** *p* < 0.001, **** *p* < 0.0001; NS, no significance. Blue arrows indicate ear capillary leakage. HFD, high-fat diet. ND, normal diet.

**Figure 2 cells-11-00779-f002:**
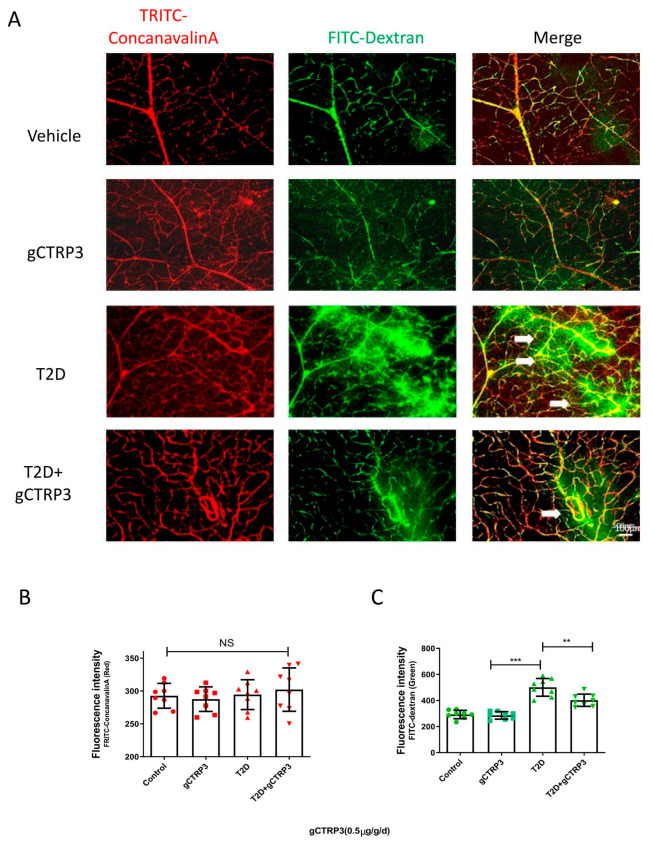
The protective effect of CTRP3 on DR. (**A**), Representative images showing that CTRP3 reduced the retinal vascular leakage, which was increased in the T2D DR group. (**B**), Representative images showed no visible neovascularization was evaluated with TRITC-ConcanavalinA signal (intact retinal vessels, red). (**C**), Significant extravasation of FITC-dextran (vascular leakage, green) was observed in T2D-DR mice, but it was reduced by CTRP3. All data are shown as means ± SD. *n* = 8–10 for each group. ** *p* < 0.01, *** *p* < 0.001; NS, not significant. White arrows indicate retinal vascular leakage. T2D, Type 2 diabetes. DR, diabetic retinopathy.

**Figure 3 cells-11-00779-f003:**
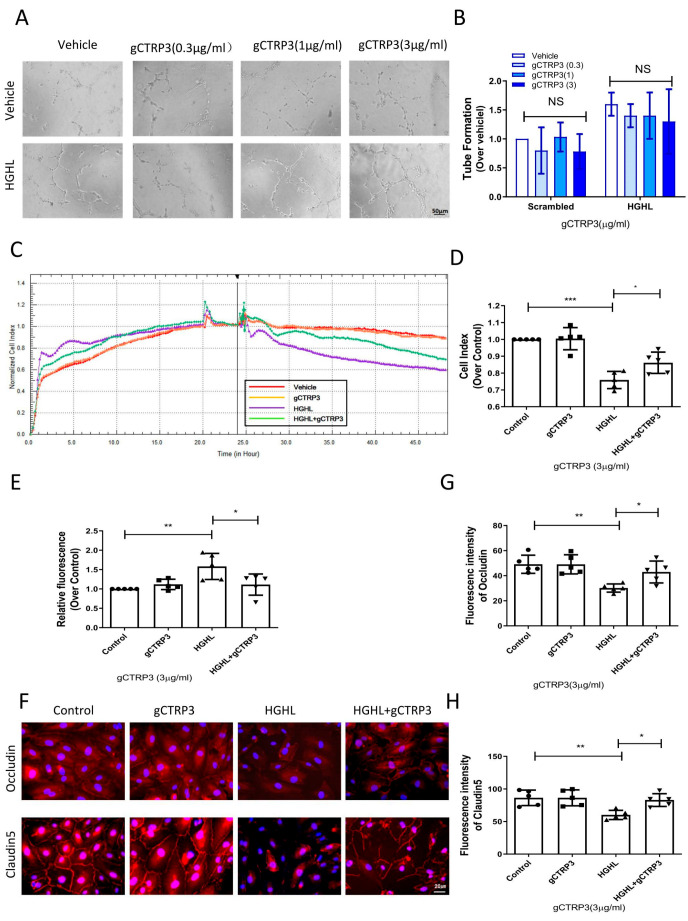
gCTRP3 protected barrier function of iBRB against HGHL-induced damage. (**A**), Representative images of tube formation assay showing that gCTRP3 failed to promote HREMC tube formation. (**B**), Bar graph for tube formation analysis, *n* = 5. (**C**), Representative images for xCELLigence electrical conductivity assays showing that gCTRP3 increased cellular connection. (**D**), Bar graph analysis for xCELLigence conductivity assay. (**E**), Bar graph analysis for trans-well endothelial permeability assay. (**F**), Immunoflurencence images showing that HGHL disrupted cell-to-cell junctions in HRMECs staining with Occludin (red), Claudin-5 (red), Nuclei (blue). (**G**,**H**), Fluorescence intensity quantification of Occludin (red), Claudin-5. HRMECs were treated with vehicle or HGHL for 24 h followed by CTRP3 administration (3 μg/mL). All data are shown as means ± SD. *n* = 5 for each group. Bar graph represents analysis from at least 5 independent repeated experiments. * *p* < 0.05, ** *p* < 0.01, *** *p* < 0.001. HGHL, high glucose, and high lipids.

**Figure 4 cells-11-00779-f004:**
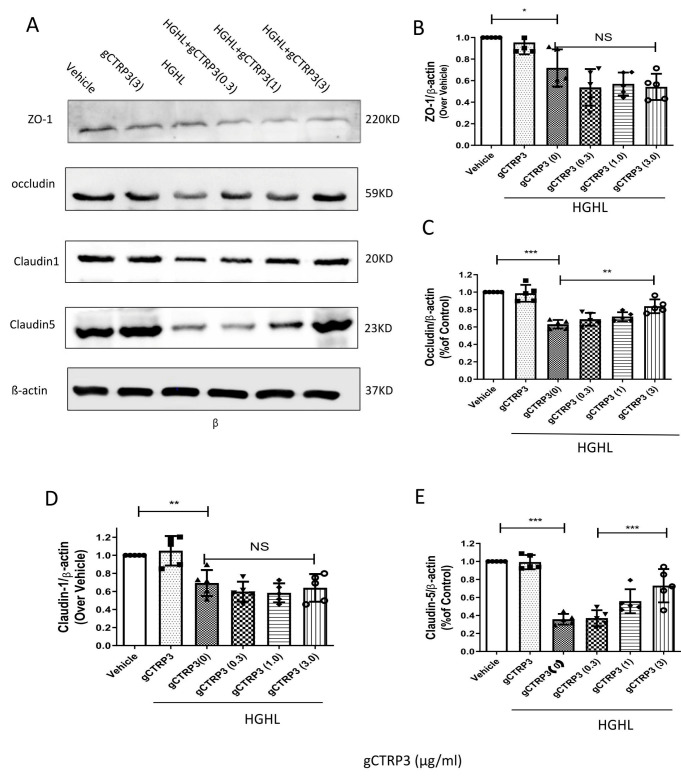
CTRP3 increased Occludin and Claudin-5 expression and protected the integrity of iBRB against HGHL-induced damage. (**A**), Representative immunoblots showing that Claudin-1, ZO-1, Claudin-5, and Occludin protein levels in HRMECs pre-treated with gCTRP3 (3 μg/mL) followed by HGHL administration. (**B**), Bar graph analysis for quantification of ZO-1 protein expression. (**C**), Bar graph analysis for the quantification of Occludin protein expression. (**D**), Bar graph analysis for quantification of Claudin-1 protein expression. (**E**), Bar graph analysis for quantification of Claudin-5 protein expression. All data are shown as means ± SD. *n* = 5–7. Bar graph represent analysis from at least 5 independent repeated experiments. * *p* < 0.05, ** *p* < 0.01. *** *p* < 0.001; NS, no significance.

**Figure 5 cells-11-00779-f005:**
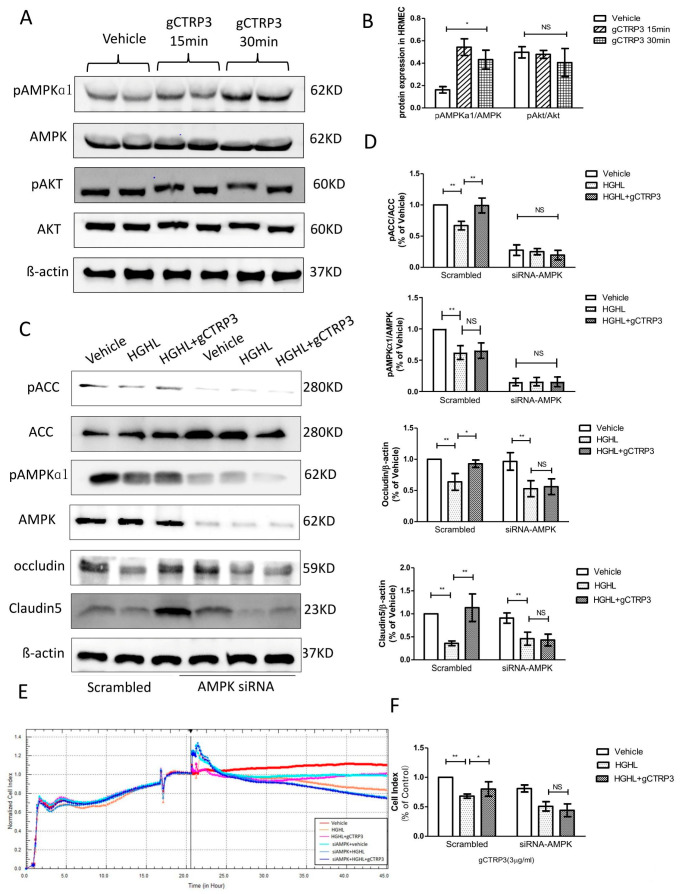
Inhibition of AMPK blocked CTRP3′s role in the upregulation of Occludin and Claudin 5. (**A**), Representative immunoblots show AMPK and Akt activation in HREMCs treated with gCTRP3. (**B**), The bar graph shows the quantification of AMPK and Akt activation. HRMECs were challenged by gCTRP3 (3 µg/mL) for 15 min followed by HGHL administration (*n* = 5–7). (**C**), Western blot analysis confirmed successful knockdown of AMPK by siRNA and expression of pAMPK, AMPK, pACC, ACC, Occludin, and Claudin-5. (**D**), Bar graph for quantification of the level of key molecules. (**E**), xCELLigence electrical conductivity assays showed the role of AMPK in HREMCs pretreated with CTRP3 followed by the HGHL challenge. (**F**), Bar graph for analysis of xCELLigence electrical conductivity assay. *n* = 5–7. Bar graph represents analysis from at least 5 independent repeated experiments. All data are shown as means ± SD. * *p* < 0.05, ** *p* < 0.01; NS, no significance. HGHL, high glucose and high lipids.

**Figure 6 cells-11-00779-f006:**
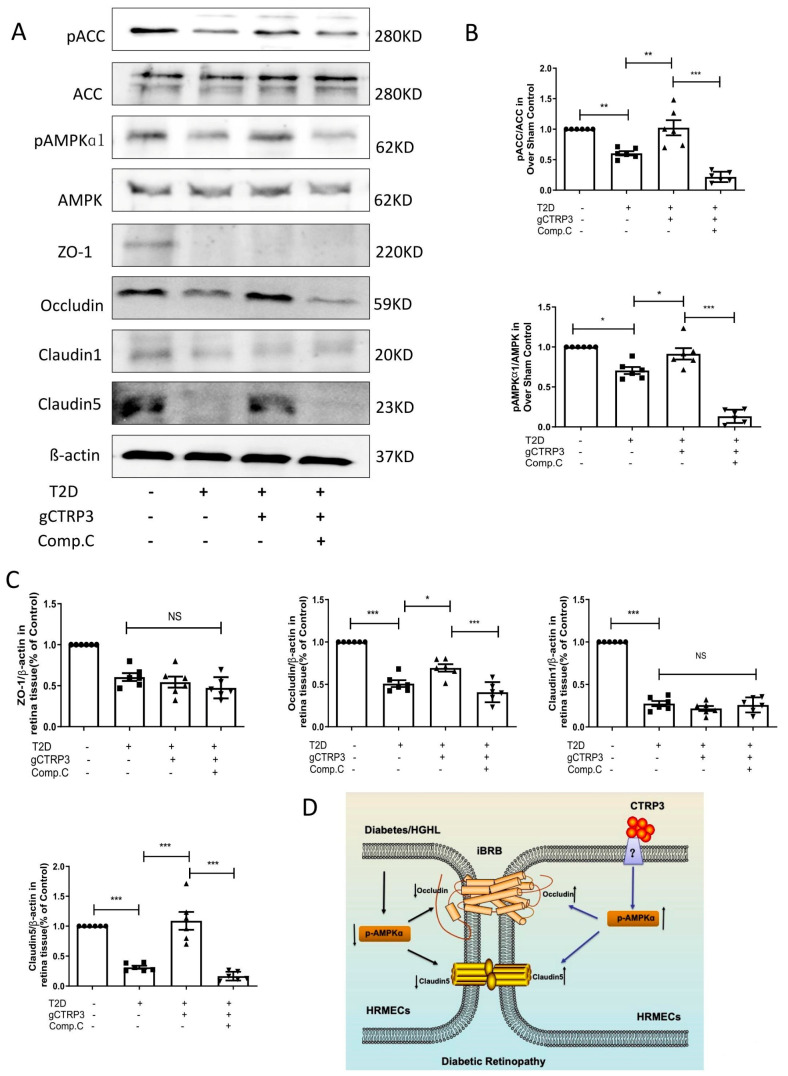
AMPK deficiency blocked CTRP3 upregulated the protein level of Occludin, Claudin-5 in DR. (**A**): Representative Western blots of pAMPK, AMPK, pACC, ACC, Claudin-1, ZO-1, Claudin-5 and Occludin in the retina from Sham and T2D/STZ DR mice with/without gCTRP3 treatment and Compound (C) administration. (**B**), Bar graph for quantification of the expression of pAMPK, AMPK, pACC, and ACC. (**C**), Bar graph for quantification of the expression of Claudin-1, ZO-1, Claudin-5, and Occludin. (**D**), Diagram depicts mechanism responsible for the iBRB protective effects of CTRP3 on diabetic retinopathy. *n* = 6–8. Bar graph represents analysis from at least 6 independent repeated experiments. Arrow pointing up indicates upregulation; Arrow pointing down indicates downregulation. All data are shown as means ± SD. * *p* < 0.05, ** *p* < 0.01. *** *p* < 0.001; NS, no significance. DR, diabetic retinopathy. T2D, Type 2 diabetes. Comp. C, Compound C.

## Data Availability

Not applicable.

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
