# Peer review of "C1q/TNF-Related Protein 3 Prevents Diabetic Retinopathy via AMPK-Dependent Stabilization of Blood–Retinal Barrier Tight Junctions"

_cells, 2022, doi:10.3390/cells11050779_

Round 1

Reviewer 1 Report

The current study by Yan et al. describes the importance of C1q/TNF-related Protein 3 in preventing DR through AMPK dependent stabilization of blood-retinal barrier tight junctions. The study is well thought-out and the results are convincing. The experimental methods are appropriate. I only have the following comments:

Line 86,87- Provide the route of administration for Evan’s blue dye and how long after injection were the mice euthanized?

Line 98,99- Were the mice perfused with PBS to remove non-adherent dye?

Line 218- typo: ‘Evens’ blue

Figure 3B- Is the X-axis labeled wrong as scrambled instead of control?

Figure 3G-Y-axis “fluorescence” is misspelled.

Figure 3F- Shows CLDN5 localization in the nucleus – Is only CTRP3 treatment induces intracellular translocation? Can the authors explain why vehicle-treated groups have CLDN5 distribution in the nucleus?

Figure 5C- Does AMPK loss in HRMECs affect CLDN5/OCCLUDIN expression in vehicle (untreated) groups? An increased CLDN5 in the AMPK siRNA vehicle group compared to the scrambled vehicle group is visible in the presented data.

There is increased tube formation in HGHL treated HRMECs – does that not mean an increase in proliferation?

Better justification on why globular domain isoform of CTRP3 was chosen for treatment?

Author Response

We greatly appreciate this reviewer’s positive assessment of our manuscript and agree with all his/her expert suggestions. We have made the following specific revisions.

Point 1. Line 86,87- Provide the route of administration for Evan’s blue dye and how long after injection were the mice euthanized?

Response1: Thank you for reminding us for this point. The mice were euthanized in 5 minutes after injection. As suggested, additional details of the study have been added to Line 88,89 of the revised version.

Point 2. Line 98,99- Were the mice perfused with PBS to remove non-adherent dye?

Response2: We followed procedures published by other investigators (reference 15-18). Mice were not perfused with PBS.

Point 3. Line 218-typo: ‘Evens’ blue

Response3: Thank you for pointing out the error. We have corrected ‘Evens’ blue to ‘Evan’s blue.

Point 4.Figure 3B- Is the X-axis labeled wrong as scrambled instead of control?
Response4: We apologize for this error, and it has been corrected in Fig3B.

Point 5. Figure 3G-Y-axis “fluorescence” is misspelled.

Response5: We apologize for this error, and it has been corrected in Fig3G.

Point 6. Figure 3F- Shows CLDN5 localization in the nucleus – Is only CTRP3 treatment induces intracellular translocation? Can the authors explain why vehicle-treated groups have CLDN5 distribution in the nucleus?
Response6: We appreciate the reviewer’s careful evaluation of our manuscript. We indeed noticed that CLDN5 florescence staining is present in the nucleus in both vehicle and CTRP9-treated group. Molecular mechanisms governing CLDN5 intracellular trafficking are likely complex and cannot be addressed in the current study. We will investigate and explain this interesting phenomenon in our future studies. We have included the discussion in line 429.

Point 7. Does AMPK loss in HRMECs affect CLDN5/OCCLUDIN expression in vehicle (untreated) groups? An increased CLDN5 in the AMPK siRNA vehicle group compared to the scrambled vehicle group is visible in the presented data.
Response7: We agreed with the reviewer that CLDN5 in AMPK siRNA vehicle group shown a tendency of elevation compared to the scrambled vehicle group. However, there is no statistical significance in CLDN5 expression between the 2 groups.

Point 8. There is increased tube formation in HGHL treated HRMECs – does that not mean an increase in proliferation?

Response8: There is an increasing tendency in tube formation at gCTRP3 1 mg/ml treatment, however, no statistical significance was identified compared to vehicle control. When treated with HGHL (HGHL, 25 mM D-glucose/250 μM palmitates) in HRMECs, tube formation was not significantly increased compared to vehicle control, which also indicates that in this setting, the proliferation did not occur.  

Point 9. Better justification on why globular domain isoform of CTRP3 was chosen for treatment?

Response9: There are 2 reasons why we have selected gCTRP3 as the treatment. First, in our previously published paper, we identified that CTRP3 downregulation in patient with diabetes is strongly associated with NR complications of diabetic patients; Second, gCTRP3 is the biologically active form of CTRP3. We apologize for lack of details in the original manuscript. The reason has been included in methods in Line 71 of the revised version and described in detail in line 362-264 and 372-374.

Reviewer 2 Report

The authors show that CTRP3 protects the inner blood-retinal barrier integrity and resists the vascular permeability induced by diabetic retinopathy. Mechanistically, the administration of CTRP3 activates AMPK signaling pathway. One major flaw is that the authors fail to display the ethics approval code number. My other comments are as below: 1. Line 428, which reference? 2. Fig 6D needs explanation in caption- e.g., arrow pointing up = upregulation? 3. Having said that, Figure 6D deserves an elaboration in the manuscript text (result or discussion section) to summarize the mechanism responsible for the iBRB protective effects of CTRP3 339 on diabetic retinopathy.

Author Response

We greatly appreciate this reviewer’s positive assessment of our manuscript and agree with all his/her expert suggestions. Specifically, we have made the following revisions. Mechanistically, the administration of CTRP3 activates AMPK signaling pathway.

Point 1. One major flaw is that the authors fail to display the ethics approval code number.

Response1: We have included the information in line 61.

Point 2. Line 428, which reference?

Response2: We apologized with the error. We have included the reference at line 434.

Point 3. Fig 6D needs explanation in caption- e.g., arrow pointing up = upregulation?

Response3: As suggested, the explanation for arrows have been added in legend of Fig6D.

Point4. Having said that, Figure 6D deserves an elaboration in the manuscript text (result or discussion section) to summarize the mechanism response

Response4: As suggested, we have included in the summary of mechanism in the discussion in line 346-348 of the revised version.